



# Campbell Plateau: A major control on the SW Pacific sector of the Southern Ocean circulation.

Aitana Forcén-Vázquez[1,2], Michael J. M. Williams[1], Melissa Bowen[3], Lionel Carter[2], and Helen Bostock[1]

[1]NIWA
[2]Victoria University of Wellington
[3]The University of Auckland

*Correspondence to:* Aitana (aitana.forcen@gmail.com)

**Abstract.** New Zealand's subantarctic region is a dynamic oceanographic zone with the Subtropical Front (STF) to the north and the Subantarctic Front (SAF) to the south. Both the fronts and their associated currents are strongly influenced by topography: the South Island of New Zealand and the Chatham Rise for the STF, and Macquarie Ridge and Campbell Plateau for the SAF. Here for the first time we present a consistent picture across the subantarctic region of the relationships between front

positions, bathymetry and water mass structure using eight high resolution oceanographic sections that span the region. Our results show that the northwest side of Campbell Plateau is comparatively warm due to a southward extension of the STF over the plateau. The SAF is steered south and east by Macquarie Ridge and Campbell Plateau, with waters originating in the SAF also found north of the plateau in the Bounty Trough. Subantarctic Mode Water (SAMW) formation is confirmed to exist south of the plateau on the northern side of the SAF in winter, while on Campbell Plateau a deep reservoir persists into the following

autumn. Antarctic Intermediate Water (AAIW) is observed in the deeper regions around the edges of the plateau, but not on the plateau, confirming that the waters on the plateau are effectively isolated from AAIW and deeper water masses that typify the open Southern Ocean waters.

## 1 Introduction

Subantarctic New Zealand has a unique bathymetry with a combination of ocean ridge and large submarine plateau that control

the extent of the Subantarctic Zone (SAZ) in the southwest Pacific sector of the Southern Ocean (Fig. 1). The SAZ lies between the Subtropical Front (STF) to the north and the Subantarctic Front (SAF) to the south (e.g., Morris et al., 2001; Neil et al., 2004; Chiswell et al., 2015). The STF is almost continuous around the Southern Hemisphere and separates the warm and salty subtropical waters transported by the subtropical gyres from the cold and fresh subantarctic waters (e.g., Deacon, 1982; Belkin and Gordon, 1996). The SAF, a circumpolar feature that isolates the southern polar regimes, marks the northern limit of the

Antarctic Circumpolar Current (ACC) (e.g., Orsi et al., 1995; Belkin and Gordon, 1996). This particularity makes the New Zealand subantarctic a key region to investigate the links between subtropical-subpolar water as both systems converge east of New Zealand at around 49°S creating a boundary between the subtropical and the subantarctic waters (e.g., Bryden and Heath, 1985; Heath, 1985; Fernandez et al., 2014).



The New Zealand subantarctic region encompasses the complex bathymetry of Macquarie Ridge, Emerald Basin, Campbell Plateau and Bounty Trough making the surrounding oceanography complex (e.g., Morris et al., 2001; Williams, 2004). Ocean circulation and the distribution of the fronts are constrained by the complex bathymetry in this area (e.g., Gordon, 1975; Heath, 1981; Williams, 2004). Studies derived from *in situ* hydrographic data suggest that bathymetry not only influences the latitude and orientation of the fronts in the ACC, but also influences its flow (Deacon, 1982; Gille, 2003; Sokolov and Rintoul, 2007).

The formation of Subantarctic Mode Water (SAMW) and Antarctic Intermediate Water (AAIW) is a primary mechanism for sequestering and storing anthropogenic $CO_2$ and chlorofluorocarbons (CFCs) within the oceans (McCartney, 1982; Sabine et al., 2002; Hartin et al., 2011; Hasson et al., 2011; Holte et al., 2012). SAMW and AAIW formation are both linked to the SAF (e.g., Hanawa and Talley, 2001; Koshlyakov and Tarakanov, 2005; Hartin et al., 2011). Mode waters are found in the New Zealand subantarctic region and hence important as an area of carbon dioxide uptake into the oceans (Currie and Hunter, 1998; Currie et al., 2011). These waters are also part of the global overturning circulation, exporting heat, salt and nutrients into the Southern Hemisphere basin-wide gyre circulation at depths isolated from the atmosphere (e.g., Herraiz-Borreguero and Rintoul, 2011; Hasson et al., 2011). They play a major role in the modulation of global climate variations by ventilating the lower thermocline of the Southern Hemisphere subtropical gyres (e.g., McCartney, 1977; Rintoul and Bullister, 1999; Sloyan and Rintoul, 2001; Morris et al., 2001).

From a regional perspective, Campbell Plateau is important for local climate, fisheries and marine mammal populations. The oceanic variability over Campbell Plateau influences New Zealand's climate by affecting the local oceanography southeast of the South Island (e.g., Hopkins et al., 2010). It also supports fisheries of economic importance that reflects the productivity associated with the fronts especially the STF (Butler et al., 1992; Murphy et al., 2001; Bradford-Grieve et al., 2003; Bradford-Grieve and Hanchet, 2002). The New Zealand subantarctic islands also support a variety of marine mammals and sea-birds whose populations are regulated by oceanographic climatic factors as well as human activities (Robertson and Chilvers, 2011; Bradford-Grieve et al., 2003). Finally, an improved understanding of this region is also important for paleoceanographic studies by providing knowledge of the processes that influence the ocean environment (e.g., Neil et al., 2004; Cortese et al., 2013).

This paper presents data from oceanographic sections collected between 1998 and 2013 (Fig. 1) to describe the spatial oceanography of the New Zealand subantarctic region, the changes in the characteristics of the STF around New Zealand and the influence of the bathymetry on the Southern Ocean fronts. Some data are in previous publications (Morris et al. (2001); Sutton (2003); Williams (2004); Griffith et al. (2010); Smith et al. (2013); Rintoul et al. (2014)), but here all the data have been reprocessed and are considered together in a consistent framework for the first time. The first part of the paper describes the oceanography of the region through hydrographic data. Frontal positions from hydrography and the alignment with the geostrophic jets are investigated and how they correspond with the bathymetry. The second part of the paper compares the hydrographic posi-



tion of the fronts with the identification of the fronts from SSH and discusses some of the unresolved oceanographic issues in this region.

**Regional setting**

The Macquarie Ridge has an average depth of $1500\,\mathrm{m}$, but contains several gaps for the SAF to pass through (e.g., Rintoul et al., 2014). These gaps are as deep as $4000\,\mathrm{m}$. Campbell Plateau is a region of submerged continental crust and extends $\sim 1000\,\mathrm{km}$ southeast from the South Island. Most of the plateau is shallower than $1000\,\mathrm{m}$, with a few areas as deep as $2000\,\mathrm{m}$. Its boundaries are well defined, with a steep continental slope that descends sharply to $4000\,\mathrm{m}$. Several subantarctic islands such as the Auckland Islands and Campbell Island are found in the region. North of Campbell Plateau, Bounty Trough is a deep failed submarine rift with an axial submarine channel system that extends $\sim 1000\,\mathrm{km}$ east from the South Island. Bounty Plateau and Bollons Seamount are its southern extreme, the latter rising from $4500\,\mathrm{m}$ to $\sim 900\,\mathrm{m}$ (Carter and Carter, 1988; Carter and Wilkin, 1999) (Fig. 1).

**Previous work**

The STF sits between 35° and 45°S, except where South America breaks the front's continuity (e.g., Deacon, 1982; Belkin and Gordon, 1996). It is a shallow front, typically extending to depths between 200 and $300\,\mathrm{m}$ and is considered the northern limit of subantarctic waters (e.g., Deacon, 1982; Orsi et al., 1995; Hamilton, 2006) and the southern boundary of the Southern Hemisphere's subtropical gyres (Smith et al., 2013). The STF has been defined as consisting of a double frontal structure, the Northern Subtropical Front (N-STF) and the Southern Subtropical Front (S-STF), that enclose the Subtropical Frontal Zone (STFZ) where STW and SAW converge (Belkin, 1988; Hamilton, 2006; Smith et al., 2013). The S-STF is also an area where micro-nutrient rich (especially iron) subtropical waters mix with the high-nutrient, low-chlorophyll subantarctic waters generating a region of high primary production (Moore and Abbott, 2000; Sutton, 2001; Murphy et al., 2001; Sokolov and Rintoul, 2002; Graham et al., 2015). South of New Zealand, only a well defined single branch is identified, with features consistent with the S-STF. This is the front's most southern position, at 49.6°S and is topographically steered as it crosses the Macquarie Ridge (Smith et al., 2013). Here, the front is density compensated, i.e., there is little or no change in density across the front resulting in little or no sea surface height (SSH) signature (Smith et al., 2013). However, it is recognized by changes in sea surface temperature (SST) or in hydrographic sections showing changes in the vertical structure of the temperature and salinity of the front (e.g., Rintoul and Bullister, 1999; Morris et al., 2001; Sokolov and Rintoul, 2002, 2007, 2009b; Smith et al., 2013). Southeast of the South Island, the S-STF density gradient increases and it is locally known as the Southland Front (SF) (e.g., Chiswell, 1996; Sutton, 2003). The associated flow, the Southland Current, transports mainly SAW close to New Zealand (Sutton, 2003). The Southland Current flows northeast steered by the continental margin (e.g., Chiswell, 1996; Sutton, 2003; Hopkins et al., 2010), until it is diverted eastwards along the southern flank of Chatham Rise and into the South Pacific Ocean (Sutton, 2001; Hopkins et al., 2010). Recent studies have have reassessed the characteristics of the STF, suggesting that there is a dynamical Subtropical Front (DSTF) which is non-density compensated (with a strong flow associated) and a STFZ - zone, which is density compensated and where the STW and SAW mix (Graham and De Boer, 2013). They suggest that the two



fronts are forced by different mechanisms and hence are different and not connected and should not be considered a continuous front.

The SAF forms the northern boundary of the ACC, and carries much of its transport (Gille, 2003; Sokolov and Rintoul, 2007).
The ACC has historically been divided in three fronts, the SAF, the Polar Front (PF) and the southern ACC front (e.g., Orsi et al., 1995; Belkin and Gordon, 1996) (Fig. 1). More detailed studies have shown that these fronts are not single static entities, but have multiple branches or filaments that change in position and intensity on different scales (Moore et al., 1999). Higher resolution data analysis has shown that the SAF consists of three branches that extend from the sea surface to the sea floor (e.g., Sokolov and Rintoul, 2007, 2009b; Rintoul et al., 2014). The positions of the three SAF branches are modified by bathymetry in the New Zealand subantarctic region. They are forced to pass through gaps in the Macquarie Ridge, and merge where the SAF impinges on Campbell Plateau (e.g., Morris et al., 2001; Sokolov et al., 2006; Rintoul et al., 2014). Downstream, the flow divides into two branches, the northern branch following the plateau bathymetry and the middle and southern branches turning sharply to the south east at $\sim 164°$E (e.g., Davis, 1998; Morris et al., 2001; Stanton and Morris, 2004; Sokolov and Rintoul, 2007). However, identification of the Southern Ocean fronts is not a straight forward task. Fronts, as water mass boundaries, can be identified by hydrographic data through strong horizontal gradients in water mass properties such as temperature, salinity and usually (e.g., Belkin and Gordon, 1996; Heath, 1985; Orsi et al., 1995; Sokolov and Rintoul, 2002; Chapman, 2014). This identification, however, becomes unclear when Campbell Plateau's bathymetry forces the different branches of the front to merge and then diverge (Sokolov and Rintoul, 2002, 2007).

Water mass identification can be a controversial issue as definitions change depending on the method used for identification (e.g., Sokolov and Rintoul, 2007; De Boer et al., 2013; Chapman, 2014). Surface waters originate locally, while intermediate and deep waters have been advected from further away, but they can be identified by their physical and chemical properties and thus, their formation process and location can be investigated (e.g., Tomczak and Godfrey, 2001; Brown et al., 2001; Talley et al., 2011). For example the identification of the surface waters between the SAF and PF. Antarctic Surface Water (AASW) lies to the south of the PF and Subantarctic Surface Water (SASW) lies to the north of the SAF. However, the surface water between the two fronts is not routinely considered a distinct water mass and has been described as "in transition" between both water masses (e.g., Carter et al., 2008; Chiswell et al., 2015). Another example is SAMW and AAIW, intermediate waters that form in the SAZ (e.g., McCartney, 1977; Morris et al., 2001; Talley et al., 2011; Bostock et al., 2013). However, the roles, presence and formation spots of these water masses on, or adjacent to, Campbell Plateau remains unclear. Whether SAMW is formed south of New Zealand has been discussed for decades: Heath (1981, 1985) suggests SAMW is formed over the plateau by deep vertical mixing. In contrast, Morris et al. (2001) proposes that SAMW found on the plateau is replenished by an inflow of dense water formed north of the SAF. Deep waters will be mentioned in the text, but are not the focus of the paper.

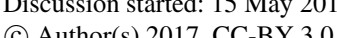



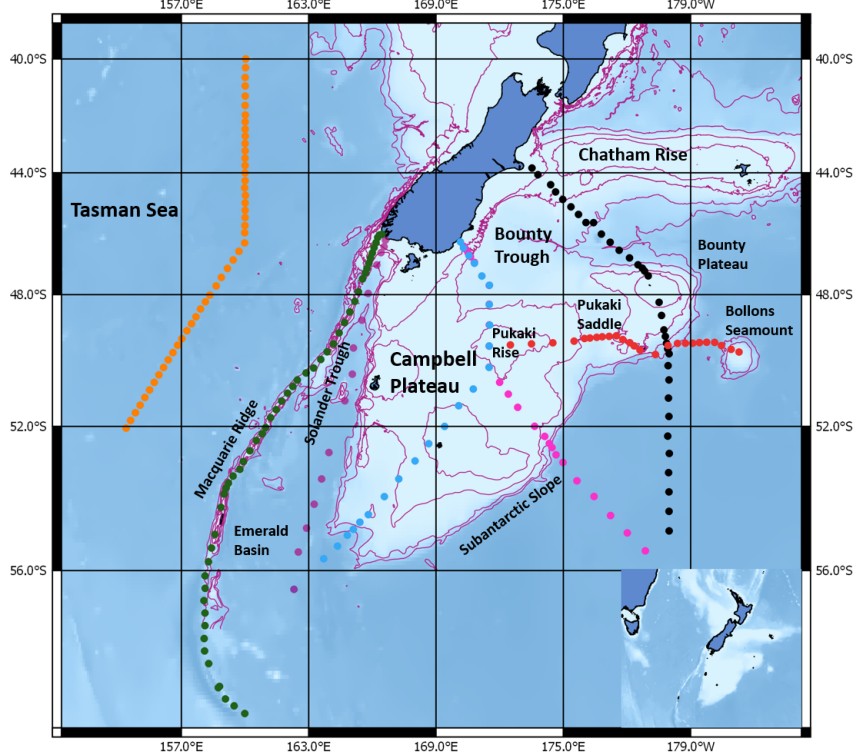

**Figure 1.** High resolution hydrographic profile positions discussed in this study. Important topographic features are shown. Details for each voyage are shown in Table 1. 500, 1000, 2000 and 3000 m bathymetric contours.

## 2 Data and Methods

Conductivity Temperature and Depth (CTD) measurements were taken on seven voyages on RV Tangaroa between 1998 and 2008. A total of 423 high resolution CTD stations were collected (Fig. 1 and Table 1), with some locations repeated on different years and different seasons. Along most of these transects there is typically increased resolution where major oceanographic features were crossed. Hydrographic profiles were conducted to near the bottom and the data at each station were averaged into 2 dbar vertical bins. When describing properties the whole water column is discussed, although depths deeper than 3000 m are not shown. All the data were re-processed from the raw files using the International Thermodynamic Equation of Seawater-2010 (TEOS-10) to calculate conservative temperature and absolute salinity (IOC et al., 2010; Pawlowicz, 2010). Absolute salinity is, as a general rule, about $0.17\,\mathrm{g\,kg^{-1}}$ saltier than practical salinity in this region. For that reason, and to be consistent with the literature, salinity measurements reported in the literature have been adjusted to absolute salinity. Conservative temperature does not need adjustment as values are the same as the potential temperature previously reported in the literature.

The transects are discussed from west to east, front identification criteria are shown in Table 2, and their positions are indicated in each transect. The transects were chosen to illustrate different characteristics of the region. The Tasman Sea transect





**Table 1.** Summary of cruises presented in this study with number of CTD casts

| Cruise name | Year | Month | Number of casts | Positions |
|---|---|---|---|---|
| TAN9806 | 1998 | 5-29 May | 68 | Solander Trough<br>Campbell Plateau |
| TAN9814 | 1998 | 4-22 Dec | 59 | Campbell Plateau<br>Bounty Trough |
| TAN9909 | 1999 | 17 Jul<br>10 Aug | 69 | Campbell Plateau<br>Bounty Trough |
| TAN0108 | 2001 | 1-21 Jun | 60 | Bounty Trough |
| TAN0307 | 2003 | 17 Apr<br>7 May | 57 | Pukaki Saddle |
| TAN0609 | 2006 | 12 Jul<br>1 Aug | 51 | Tasman Sea |
| TAN0803 | 2008 | 26 Mar<br>26 Apr | 59 | Macquarie Ridge |

shows the structure west of the subantarctic region, the Macquarie Ridge transect shows the behaviour of the fronts through the ridge, the Solander Trough transect illustrates the transition between the ridge and Campbell Plateau. The Campbell Plateau transects show: the structure of the STF south of New Zealand in the northern part of the transects; the oceanographic structure on the southwest edge of Campbell Plateau; and the effect of the plateau on the SAF. The Bounty Trough transect shows the structure to the north of the subantarctic region, while the Pukaki Saddle transect completes the general view by showing a likely path for SAF waters to reach Bounty Trough (Fig. 1).

The STF in the south Tasman Sea is identified by its double structure, the North Subtropical Front (N-STF) and the South Subtropical Front (S-STF) that encompass an eddy-rich zone with little frontal structure with the entire STF being around 300 km wide (Hamilton, 2006; Smith et al., 2013). The S-STF in the New Zealand region is found where the temperature gradient decreases from 12 °C to 10 °C at a depth of 100 to 200 m Table 2. The front is also characterized by a high salinity tongue, with a range of 34.77 g kg$^{-1}$ to 35.17 g kg$^{-1}$ (e.g., Smith et al., 2013). The STF is limited to the upper 500 m of the water column (Belkin and Gordon, 1996; Sokolov and Rintoul, 2002; Smith et al., 2013).

The SAF can be identified by strong gradients in properties that extend from the sea surface to the sea floor. It can have up to three different branches each of which correspond to a maxima in properties. The northern branch (SAF-N) can be identified as a decrease from >8 °C to <6 °C at a depth of 300 to 400 m and a salinity of 34.37 g kg$^{-1}$ to 34.77 g kg$^{-1}$. The middle branch (SAF-M) is characterised by temperatures of >6 °C to <5 °C and the southern branch (SAF-S) is where the temperature decreases from >5 °C to <3 °C at the same depths (300 to 400 m) (e.g., Sokolov and Rintoul, 2002; Sokolov et al., 2006).

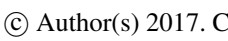



The northern branch of the Polar Front (PF) was crossed by some of the transects included in this work. Sokolov and Rin-
toul (2002) identified the PF as a double front structure in SR3 to the east of this region at 140°E. In their work, the northern
branch of the PF is a steady feature found between 53.0° to 54.3°S, and it can be identified by its subsurface temperature
minimum signature, approximating the 2 °C isotherm at around 200 m (Botnikov, 1963; Belkin and Gordon, 1996; Sokolov
and Rintoul, 2009a).

**Table 2.** Fronts identification criteria. South Subtropical Front (S-STF); Southland Front (SF); Subantarctic Front (SAF-N, -M, and -S denote
the northern, middle, and southern branches, respectively); Polar Front (PF).

| Front | Criteria | References |
|---|---|---|
| S-STF | Temperature decrease from >12 to <10 at 100-200m depth. High salinity tongue of 34.77 to 35.17 | Smith et al. (2013); Hopkins et al. (2010) |
| SF | | Sutton (2003) |
| SAF-N | Temperature decrease from >8 to <6 at 400m | Sokolov and Rintoul (2002) |
| SAF-M | Temperature decrease from >6 to <5 at 400m | Sokolov and Rintoul (2002) |
| SAF-S | Temperature decrease from >5 to <3 at 400m | Sokolov and Rintoul (2002) |
| PF | <2 at z<200m further south | Botnikov (1963) |

Geostrophic velocities were calculated from the density gradients using the geostrophic method. Geostrophic velocities
are perpendicular to the CTD sections and relative to the deepest common depths between adjacent station pairs, assuming
that depth as the "level of no motion". Cumulative transport was calculated by integrating the transport along the whole wa-
10 ter column, setting the zero transport at the stations closer to the coast. For water mass identification $\Theta$-S$_A$ (conservative
temperature-absolute salinity) diagrams are used for each transect. A compilation of the water masses definitions from the
literature is found in Table 3.

Fronts defined by hydrography on each transect are compared with mean historical positions. The mean position of the STF is
15 from Orsi et al. (1995) and it is defined by the decrease from 12 °C to 10 °C at 100 m, and its surface signature being warmer
than 11.5 °C. The mean position of the Southern Ocean fronts are defined by SSH and are from Sokolov and Rintoul (2009a).
They used a Mean Dynamic Topography (MDT) relative to 2500 dbar to define the fronts' positions. This level has been used
before in this area as the fronts are steered by bathymetry shallower than 2000 dbar (Cotroneo et al., 2013).



**Table 3.** Subantarctic region water mass properties. Antarctic Intermediate Water (AAIW); Antarctic Surface Water (AASW); Circumpolar Surface Water (CSW); Lower Circumpolar Deep Water (LCDW); Neritic Water (NW); Subantarctic Mode Water (SAMW); Subantarctic Surface Water (SASW); Subtropical Water (STW); Upper Circumpolar Deep Water (UCDW)

| | Temperature [C] | Absolute salinity | Density [kg m-3] | Depth [m] | Other features | References |
|---|---|---|---|---|---|---|
| AAIW | 4 to 8 | 34.45-34.57 | 27.1 | 600-1200 | Salinity minimum core | Bostock et al. (2013) |
| AASW | -1.9 to 1 winter<br>-1 to 4 summer | 33.16-34.66 | | less than 50m<br>thickness in the summer | south of the PF | Orsi et al. (1995) |
| CSW | 5 to 8 | 34.66 | | | Salinity drop | Morris et al. (2001)<br>Carter et al. (2008) |
| LCDW | | 34.88 | 28.27 lower<br>boundary | 2500-3000 | salinity maximum<br>derived from NADW | Sokolov and Rintoul (2002)<br>Talley et al. (2011) |
| NW | >13 | 34.56 to 34.76 | | | close to the coast<br>STW modified by freshwater outflow | Butler et al. (1992)<br>Smith et al. (2013) |
| NADW | 2 to 3 | 34.86-35.06 | 1500-3000 | | | Pickard and Emery (1990) |
| SAMW | 7<br>over the plateau | 34.51-34.66<br>over the plateau | 26.80 - 27.06 | 200-500<br>over the plateau | weak stratification<br>minimum in<br>potential vorticity | McCartney (1977)<br>Tomczak and Godfrey (2001) |
| SASW | 4 to 10 winter<br>14 summer | 34.06-34.16 summer<br>33.16 winter | | up to 500m | north of SAF | Talley et al. (2011) |
| STW | | high salinity core | | 0-200 | north of 45S | Sokolov and Rintoul (2002) |
| UCDW | 2 to 2.5 | 34.86 | 1500 | | below AASW and AAIW | Sokolov and Rintoul (2002)<br>Carter et al. (2008) |

# 3 Results

## 3.1 Hydrographic front identification and water mass analysis.

**Tasman Sea transect**

The stations of the Tasman transect, from 40°S to 52°S, were occupied in July 2006 during TAN0609 (winter). The S-STF is identified by a surface change in temperature from 11 °C to 10 °C at 45.6°S, with a corresponding signal in salinity shown as a tongue of saline STW ($35.2\,\mathrm{g\,kg^{-1}}$) protruding poleward (Table 3) (Fig. 2a and b). The tongue evolves into a weakly stratified layer in the upper $500\,\mathrm{m}$ of the water column, with a temperature range between 7 and 9 °C. It lies between 47°S and 52°S and has characteristics of SAMW (Table 3). Further south, the isotherms tilt approaching the SAF. From $500\,\mathrm{m}$ to $1500\,\mathrm{m}$ depth there is a stratified layer (AAIW), while below this the temperature decreases uniformly. At the southern end of the transect, a zone of AAIW formation is seen in the salinity transect as a fresher tongue of water from the surface deepening towards the $1250\,\mathrm{m}$ and protruding north until 43°S (Fig. 2b). Below AAIW, a layer of uniform temperature and salinity UCDW is observed. LCDW is not seen in the section because it is below $3000\,\mathrm{m}$. The $\Theta$-$S_A$ diagram shows the different water masses on the transect (Table 3), from warm and saline STW at the surface, to LCDW, with the characteristic salinity maximum core (Fig. 2e). The velocity field is complex with continuous recirculation along the entire transect constrained to the first $1500\,\mathrm{m}$


depth. A strong jet ($>25\,\mathrm{cm\,s^{-1}}$) associated with the SAF is seen at the southern extreme of the transect (Fig. 2c) coinciding with a peak in transport (Fig. 2d).

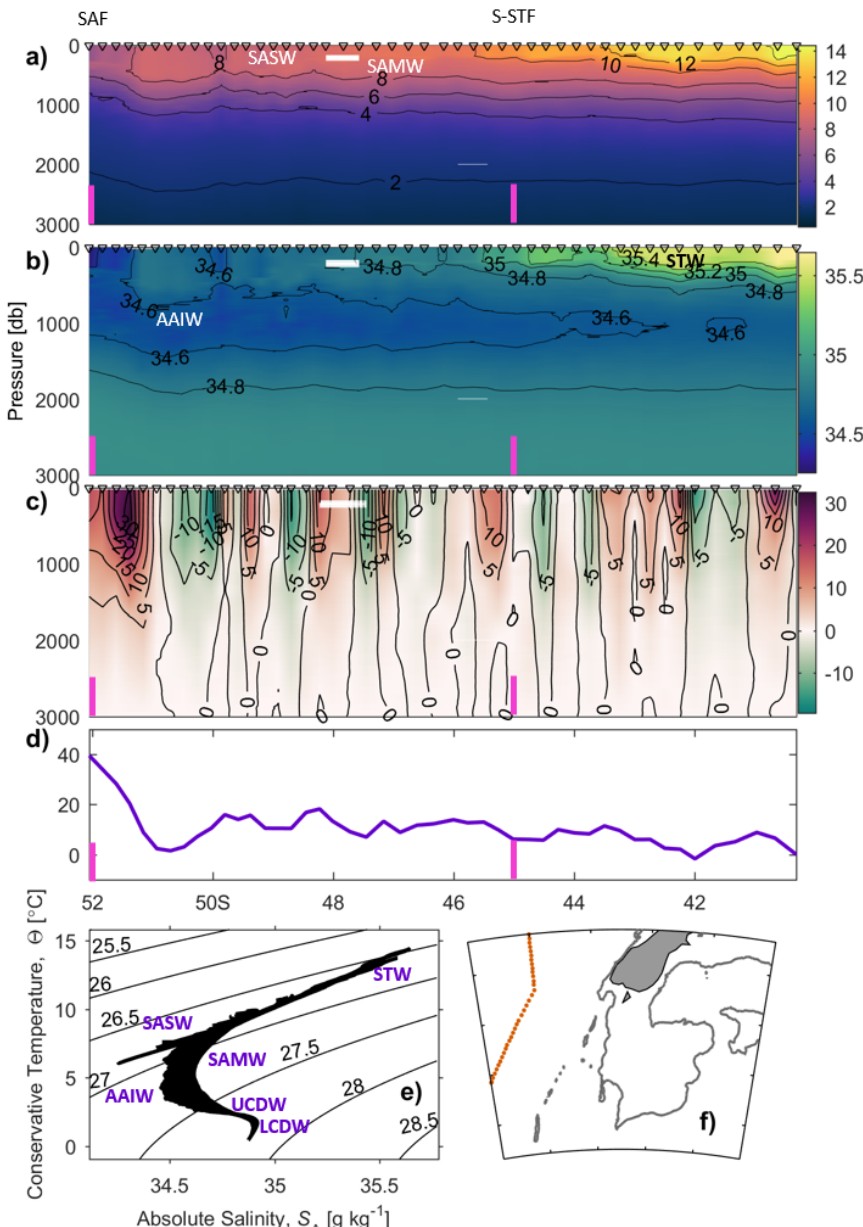

**Figure 2.** Tasman Sea: a) conservative temperature, contours every $2\,^\circ\mathrm{C}$, b) absolute salinity, contours every $0.2\,\mathrm{g\,kg^{-1}}$, c) geostrophic velocity, contours every $5\,\mathrm{cm\,s^{-1}}$ positive values to the east, d) cumulative transport (Sv), e) $\Theta$-$S_A$ diagram and f) transect position. Positions of the stations are marked on the top of the figure, fronts are marked with a magenta bar. Bathymetric features indicated on Fig. 1.



**Macquarie Ridge transect**

The stations of the Macquarie Ridge transect start at the southern tip of the South Island and follow the Macquarie Ridge to 60°S, and were occupied in April during TAN0803 (autumn) (Fig. 3f). The S-STF was crossed at 49.6°S, the same position as reported by Smith et al. (2013). It is identified by the subsurface change in temperature from >12 °C to <10 °C at a depth of

5 200 m and the disappearance of the salinity tongue (Fig. 3a and b). The temperature and salinity gradients of the STF here are largely density compensated and the associated geostrophic flow is weak (Fig. 3c). However, there is a recirculation pattern with eastward flow near the coast and westward flow at around 47.75°S. Three branches of the SAF were crossed in this transect passing through the gaps in the ridge (Fig. 3a and b). The northern branch is found at approximately 53°S, the middle branch at approximately 53.3°S, while the SAF-S is poorly defined at 56°S. The PF is identified by a subsurface temperature minimum

of 2 °C at 200 m depth at 57°S (Fig. 3a), and the deepening of the isopycnals delimiting the front (Fig. 3b). Along Macquarie Ridge currents are intensified through the ridge gaps, with a peak in transport (Fig. 3c and d). A range of surface water masses are found in this region, from Neritic Waters (NW) close to the coast to AASW south of the PF. However, intermediate waters are not well identified in this region, with a very small signal in the $\Theta$-S$_A$ diagram. What could be SAMW is found as a small patch just north of the SAF-N, and a tongue with characteristics of AAIW is found underneath (Fig. 3e). Deep waters include

UCDW shown in the transect, LCDW is not seen in the section because it is below 3000 m, but it is evident in the $\Theta$-S$_A$ diagram (Fig. 3e).




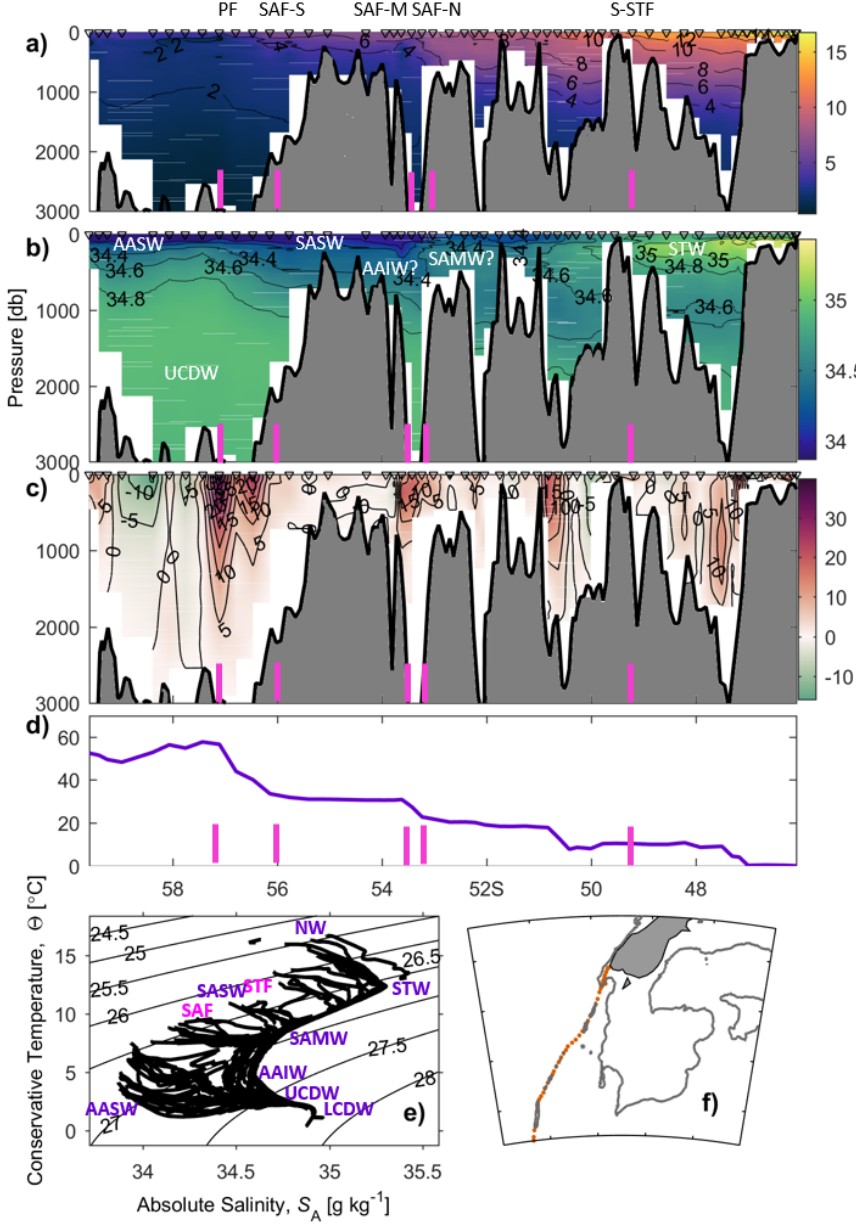

**Figure 3.** Macquarie Ridge: a) conservative temperature, contours every $2\,°C$, b) absolute salinity, contours every $0.2\,\mathrm{g\,kg^{-1}}$, c) geostrophic velocity, contours every $5\,\mathrm{cm\,s^{-1}}$ positive values to the east, d) cumulative transport (Sv), e) $\Theta$-$S_A$ diagram and f) transect position. Positions of the stations are marked on the top of the figure, fronts are marked with a magenta bar. Bathymetric features indicated on Fig. 1.





**Solander Trough transect**

The Solander Trough transect towards the Emerald Basin was carried out in May during TAN9806, (late autumn). The S-STF is better identified by the subsurface high salinity tongue ($>34.8\,\mathrm{g\,kg^{-1}}$) protruding south in the Solander Trough and the $10\,°\mathrm{C}$ isothermal outcropping to the surface (Fig. 4b and a). The front here is density compensated, hence there is no flow or transport

5  associated with it (Fig. 4c and d). Further south the oceanography is dominated by the SAF. Branches of the SAF were crossed further south compared to the Macquarie Ridge transect and are closer together, probably due to bathymetric forcing, making it difficult to separate them into independent branches. The SAF-N is crossed at $54°$S, followed by the SAF-M at $54.2°$S and the SAF-S at around $56°$S. A strong flow associated with the SAF $>40\,\mathrm{cm\,s^{-1}}$ dominates the velocity field with the hint of a second jet on the southern extreme of the transect, probably associated with the PF (Fig. 4c). These features are also seen

10  in the transport section (Fig. 4d). NW is found at the northern end of the transect close to the coast, while cool, fresh water is encountered south of the SAF, most likely a transition between AASW and SASW. Evidence of SAMW is seen at $400\,\mathrm{m}$. AAIW is present between $900\,\mathrm{m}$ and $1200\,\mathrm{m}$, with UCDW below this, down to $3000\,\mathrm{m}$ depth. LDCW is seen below $3000\,\mathrm{m}$ in the $\Theta$-S$_A$ diagram (Fig. 4e).



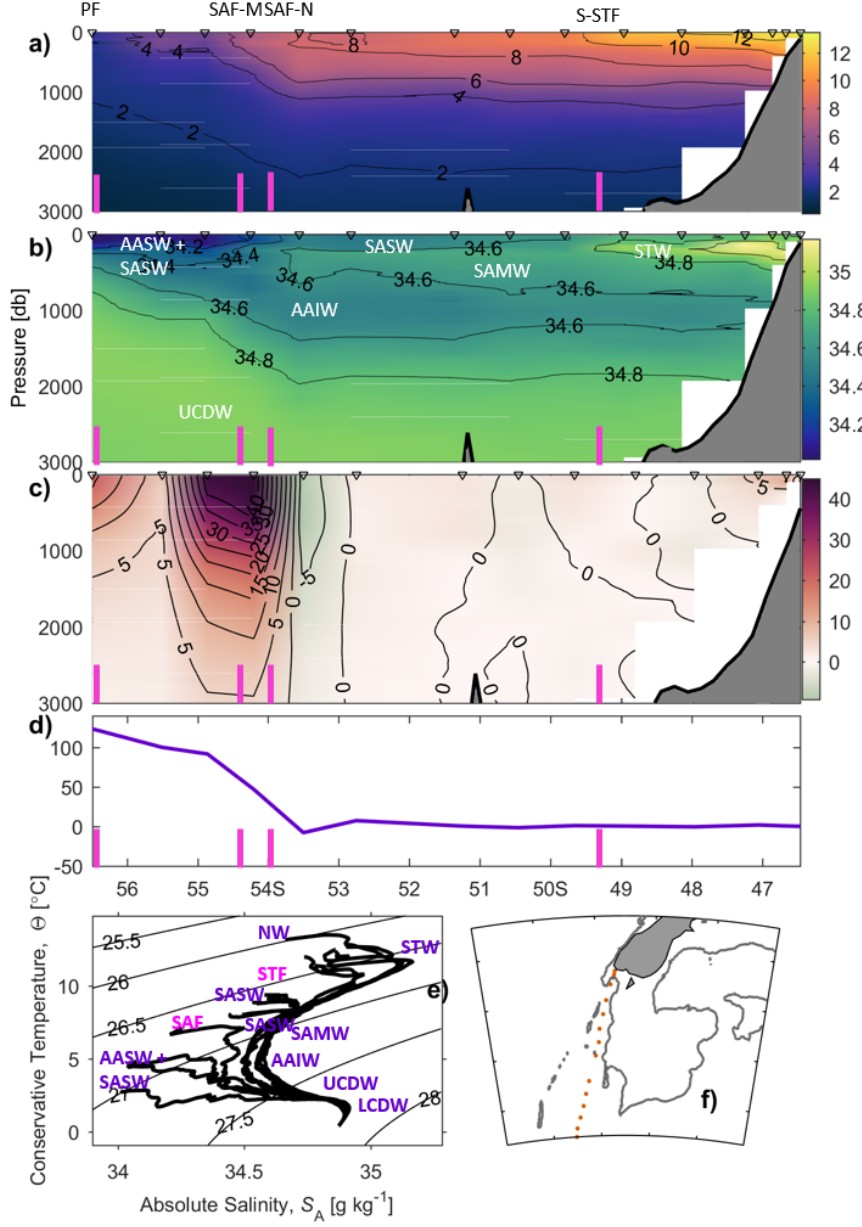

**Figure 4.** Solander Trough: a) conservative temperature, contours every $2\,°C$, b) absolute salinity, contours every $0.2\,\mathrm{g\,kg^{-1}}$, c) geostrophic velocity, contours every $5\,\mathrm{cm\,s^{-1}}$ positive values to the east, d) cumulative transport (Sv), e) $\Theta$-$S_A$ diagram and f) transect position. Positions of the stations are marked on the top of the figure, fronts are marked with a magenta bar. Bathymetric features indicated on Fig. 1.





### Western Campbell Plateau transect

Western Campbell Plateau transect was taken in May during TAN9806 (late Autumn). The northern side of the transect shows the STF through its local expression as the Southland Front (e.g., Chiswell, 1996; Sutton, 2003), and its associated strong horizontal gradients in temperature and salinity (Fig. 5a and b), with salinities >34.7 g kg$^{-1}$ on the coastal side of the transect.

There is no subsurface high salinity tongue, as seen in the previous transects (Fig. 5b). The Southland Front reaches approximately 400 m deep. In this area, the front changes into a dynamical front, hence a strong north-eastward flow close to the coast (10 cm s$^{-1}$) is present, a signal is also present in the transport (Fig. 5c and d). However, the core of the jet appears to be south of the front, not aligned with it and with a strong recirculation further south. SAMW occupies subsurface Campbell Plateau overlaid by a layer of SASW. The southern part of the transect is dominated by the SAF, with isohalines outcropping

to the surface (Fig. 5a, b and c). The distinction between the northern, middle and southern branch of the SAF is unclear in this transect, this is likely due to the fact that the different branches are merged due to the bathymetry of Campbell Plateau. In this case, the front is identified from the velocity field with a strong jet >50 cm s$^{-1}$ from the surface through the entire water column and the corresponding transport (Fig. 5c and d). The PF was crossed on the southern end of the transect, being better defined in the transport signal than in temperature or salinity (Fig. 5d). Water masses in this end of the transect range from a

transition between AASW and SASW south of the SAF, AAIW occupying the mid water layers and UCDW below the AAIW. LDCW is not seen in the transect as is below 3000 m, but isevident on the Θ-S$_A$ diagram (Fig. 5e).




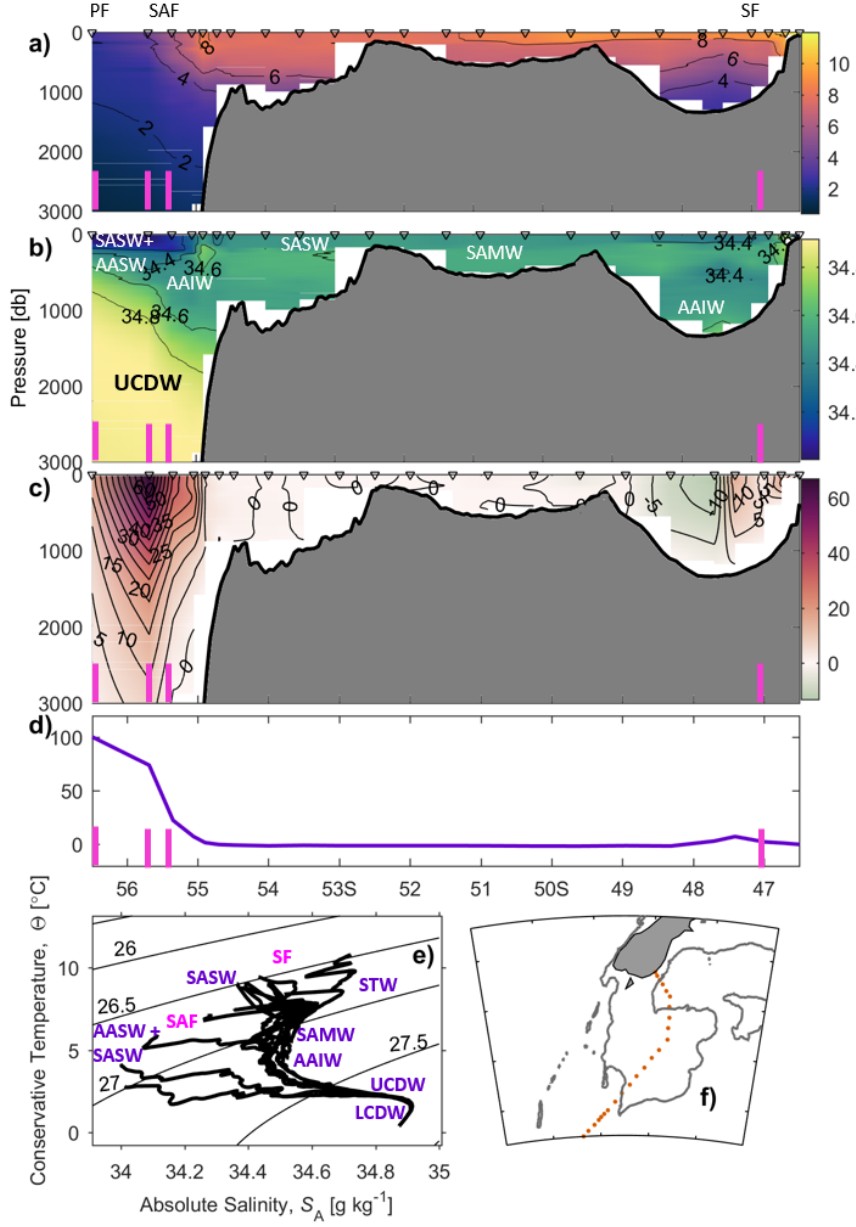

**Figure 5.** Western Campbell Plateau: a) conservative temperature, contours every $2\,°\text{C}$, b) absolute salinity, contours every $0.2\,\text{g kg}^{-1}$, c) geostrophic velocity, contours every $5\,\text{cm s}^{-1}$ positive values to the east, d) cumulative transport (Sv), e) $\Theta$-$S_A$ diagram and f) transect position. Positions of the stations are marked on the top of the figure, fronts are marked with a magenta bar. Bathymetric features indicated on Fig. 1. Note Southland Front (SF).





**Eastern Campbell Plateau transect**

The transect on the eastern side of Campbell Plateau was taken in August during TAN9909 (late winter), going from the southeast of the New Zealand coast, across Campbell Plateau and crossing Subantarctic Slope (Fig. 6f). The Southland Front (SF) is identified close to the coast by strong horizontal gradients in temperature and salinity, and no subsurface high salinity tongue (Fig. 6a and b). As a dynamical front, it has a strong flow associated it, and similar to the previous section, the jet is to the south of the actual front so is the transport (Fig. 6c and d). The surface waters are modified STW close to the coast and a homogeneous layer of SAMW sits on Campbell Plateau. Across the Subantarctic Slope only the northern branch of the SAF can be identified following the bathymetry of the plateau, with the isotherms outcropping at the surface (Fig. 6a and b). A strong jet $>70\,\mathrm{cm\,s}^{-1}$ throughout the whole water column is associated with the front (Fig. 6d). This jet contrasts with the previous analysis of this transect by Stanton and Morris (2004), who showed a double velocity core structure that is not evident here. At the surface south of the SAF, a transition between SASW and AAWS is found and below a layer of AAIW. Further down the water column sit layers of UCDW, while LCDW is seen in the $\Theta$-$S_A$ diagram (Fig. 6e).



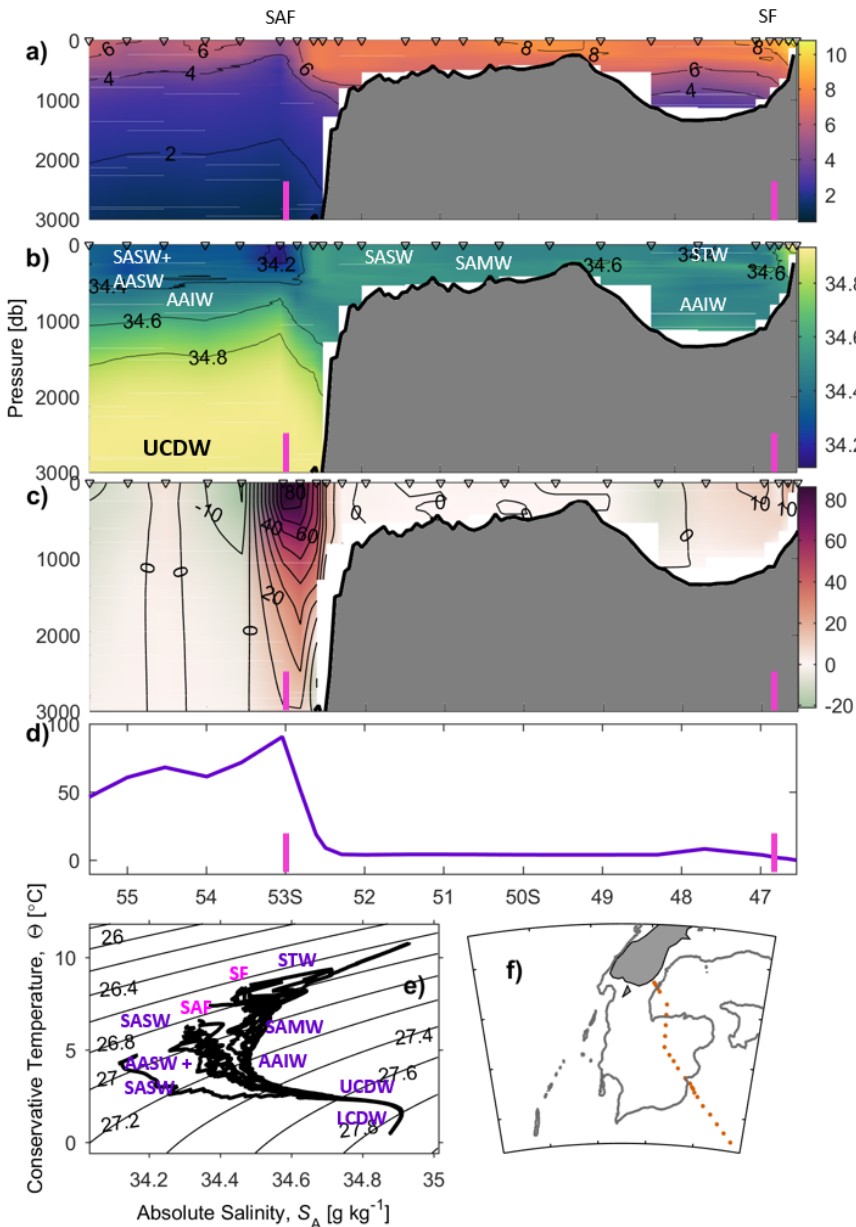

**Figure 6.** Eastern Campbell Plateau: a) conservative temperature, contours every $2\,^{\circ}$C, b) absolute salinity, contours every $0.2\,\mathrm{g\,kg}^{-1}$, c) geostrophic velocity, contours every $10\,\mathrm{cm\,s}^{-1}$ positive values to the east, d) cumulative transport (Sv), e) $\Theta$-$S_A$ diagram and f) transect position. Positions of the stations are marked on the top of the figure, fronts are marked with a magenta bar. Bathymetric features indicated on Fig. 1. Note Southland Front (SF).





**Bounty Trough transect**

The Bounty Trough transect was carried out in August during TAN9909 (late winter), and runs southeast across the Bounty Trough from the east coast of the South Island, across the Bounty Plateau and south into the SW Pacific Basin (Fig. 7f). At the northern extreme of the transect the Southland Front is close to the continental margin and is shown by strong horizontal

5   gradients in temperature and salinity (Fig. 7a and b). A tongue of high salinity water (>34.6 g kg$^{-1}$) extends from the surface to approximately 500 m (Fig. 7b), similar to that reported by Chiswell (1996). The water inshore of the front is STW and the offshore water masses are closer to SASW. A complex circulation associated with the front is observed in Bounty Trough with alternating flows that extend through the whole water column and result in a weak net transport (Fig. 7d and e). South of Bounty Plateau one branch of the SAF is identified following the bathymetry as a strong gradient in temperature and salinity

10  (Fig. 7a and b), which is also revealed on the velocity field as a strong jet >40 cm s$^{-1}$ and a peak in transport (Fig. 7c). At the southern extreme of the transect a cold core eddy is identified, at 54°S, which creates a recirculation and a large fluctuation in the transport (Figure 7a, b, c and e). AAIW is found from 900 m to 1200 m depth, while UCDW is present at the base of the water column on both sides of Bounty Plateau (Fig. 7a, b and e). LCDW is also present in deeper stations below 3000 m only seen in the Θ-S$_A$ diagram (Fig. 7e).




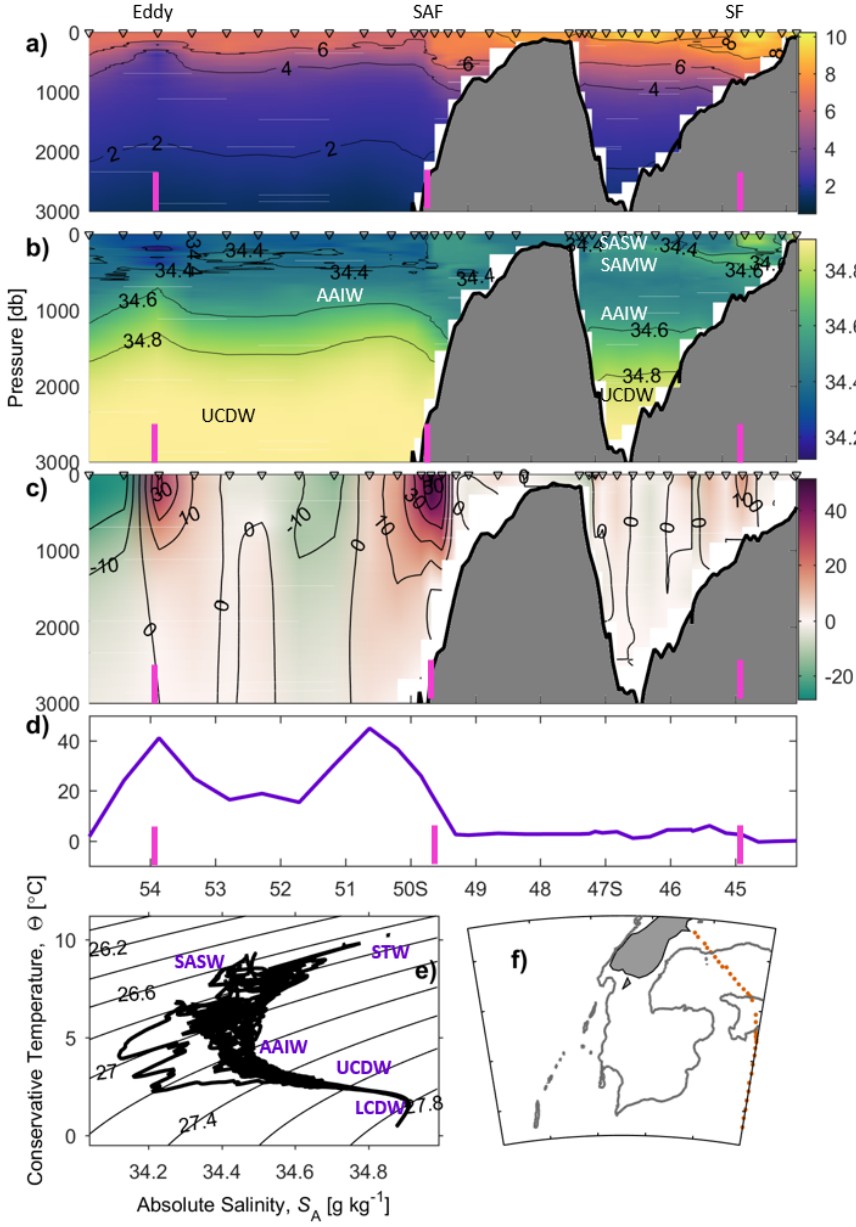

**Figure 7.** Bounty Trough: a) conservative temperature, contours every $2\,°\mathrm{C}$, b) absolute salinity, contours every $0.2\,\mathrm{g\,kg^{-1}}$, c) geostrophic velocity, contours every $10\,\mathrm{cm\,s^{-1}}$ positive values to the east, d) cumulative transport (Sv), e) $\Theta$-$S_A$ diagram and f) transect position. Positions of the stations are marked on the top of the figure, fronts are marked with a magenta bar. Bathymetric features indicated on Fig. 1. Note Southland Front (SF).





**Pukaki Saddle transect**

The Pukaki transect crosses Pukaki Saddle towards Bollons Seamount and it was carried out in May during TAN0307 (late Autumn) (Fig. 8f). This transect shows a very interesting structure due to the bathymetry it crosses and the position of the SAF in this region. The eastern part of the section sits above the Campbell Plateau and Pukaki Rise, with warm SASW at the top of

5 the water column ($10\,°C$), below that a thick homogeneous layer of SAMW occupies the rest of the water column (Fig. 8a and b). The next part crosses Pukaki Saddle, with a layer of cold and fresh SASW occupying the surface of the water column ($8\,°C$; $34.4\,\mathrm{g\,kg^{-1}}$) and a subsurface salinity maximum underneath. This gap is likely the gateway for the SASW coming from the SAF to recirculate to the northern edge of Campbell Plateau ($176.2°E$ to $178.2°W$). Below, SAMW occupies the subsurface ($100$ to $400\,m$) and AAIW fills the rest of the water column. The transect then continues towards Bollons Seamount through

depths of over $4000\,m$. This part of the transect is heavily influenced by the meandering of the SAF in this region. The transect crosses the front twice resulting in a complex structure, a cold and fresh pool of SASW at the subsurface ($5\,°C$; $34.4\,\mathrm{g\,kg^{-1}}$), and a double core jet in the velocity field flowing northward when the front is crossed once, with another jet flowing southwards when the front is crossed for the second time (Fig. 8c and d). Finally the warmer and saltier water found on the eastern end of the transect is part of a warm eddy structure reported by Williams (2004) likely influenced by the STF. AAIW is found across

the whole transect at $1200\,m$. On the eastern side of the transect, UCDW is found underneath AAIW. LCDW is found below $3000\,m$ in the deeper stations only seen in the $\Theta$-$S_A$ diagram (Fig. 8e).



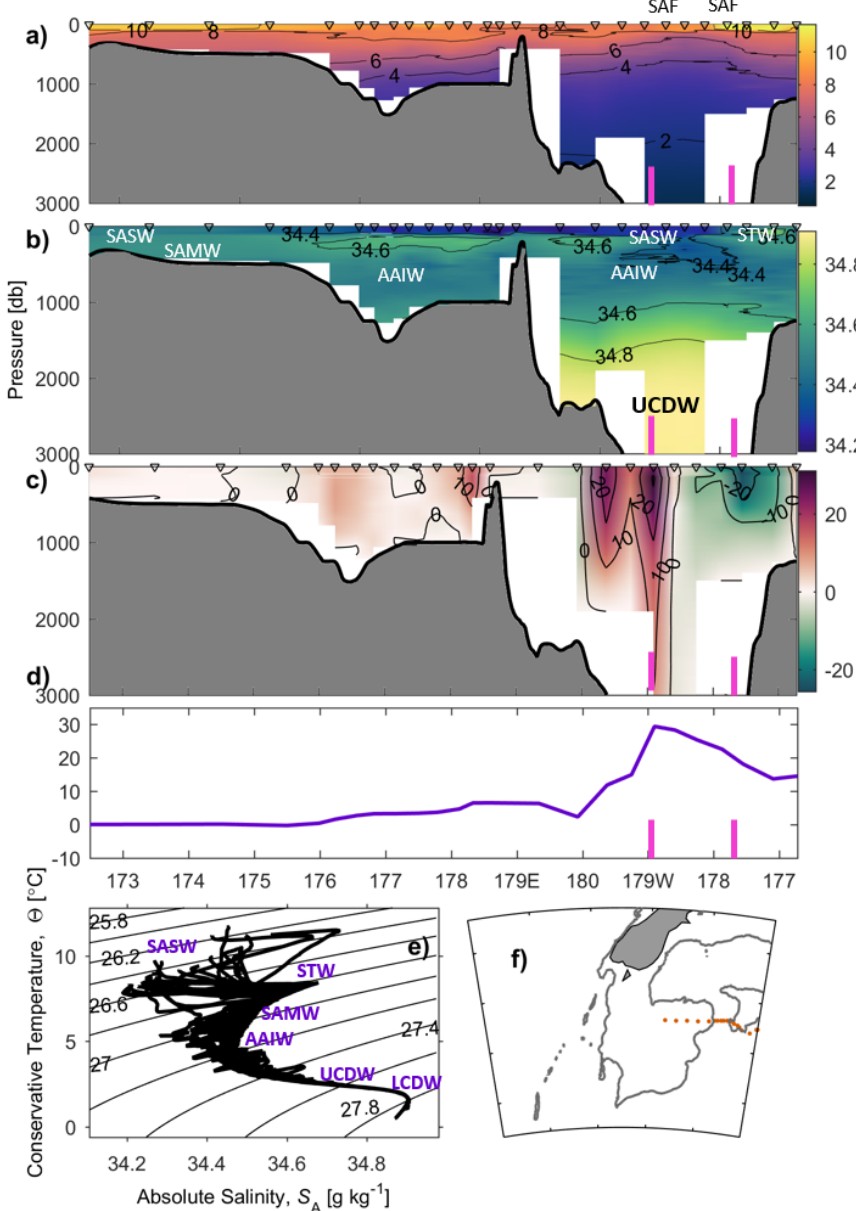

**Figure 8.** Pukaki Saddle: a) conservative temperature, contours every $2\,°C$, b) absolute salinity, contours every $0.2\,\mathrm{g\,kg^{-1}}$, c) geostrophic velocity, contours every $10\,\mathrm{cm\,s^{-1}}$ positive values to the east, d) cumulative transport (Sv), e) $\Theta$-$S_A$ diagram and f) transect position. Positions of the stations are marked on the top of the figure, fronts are marked with a magenta bar. Bathymetric features indicated on Fig. 1.



## 3.2 Hydrographic fronts vs Sea Surface Height fronts

Fig. 9 shows the position of the fronts determined along each of the transects in comparison to the published positions of the fronts; STF location from Orsi et al. (1995) defined through hydrography and the SAF, PF fronts are from Sokolov and Rintoul (2009a) using satellite SSH (these are mean positions of the fronts). There is significant disagreement between the positions

5  identified by Orsi et al. (1995) and those found in this study which is expected as they used fewer transects to define the front. We have found the STF further south, except in the Tasman Sea, where according to our study the STF sits further north. In contrast the SAF and PF fronts show broad agreement between the two methods. The exceptions are in the Macquarie Ridge, Solander Trough and Western Campbell Plateau transects, where the SAF-N is found further south in the hydrography. The reason of this disagreement is likely due to the modification of the fronts' position by the steep bathymetry of Campbell Plateau,

10  which is not resolved by either the hydrography nor the satellite data. However, where the bathymetry is not forcing the front's location, agreement is better. For example, comparison between methods highlights the double crossing of the SAF-N in the Pukaki Saddle transect and its meander between Campbell Plateau and Bollons Seamount.




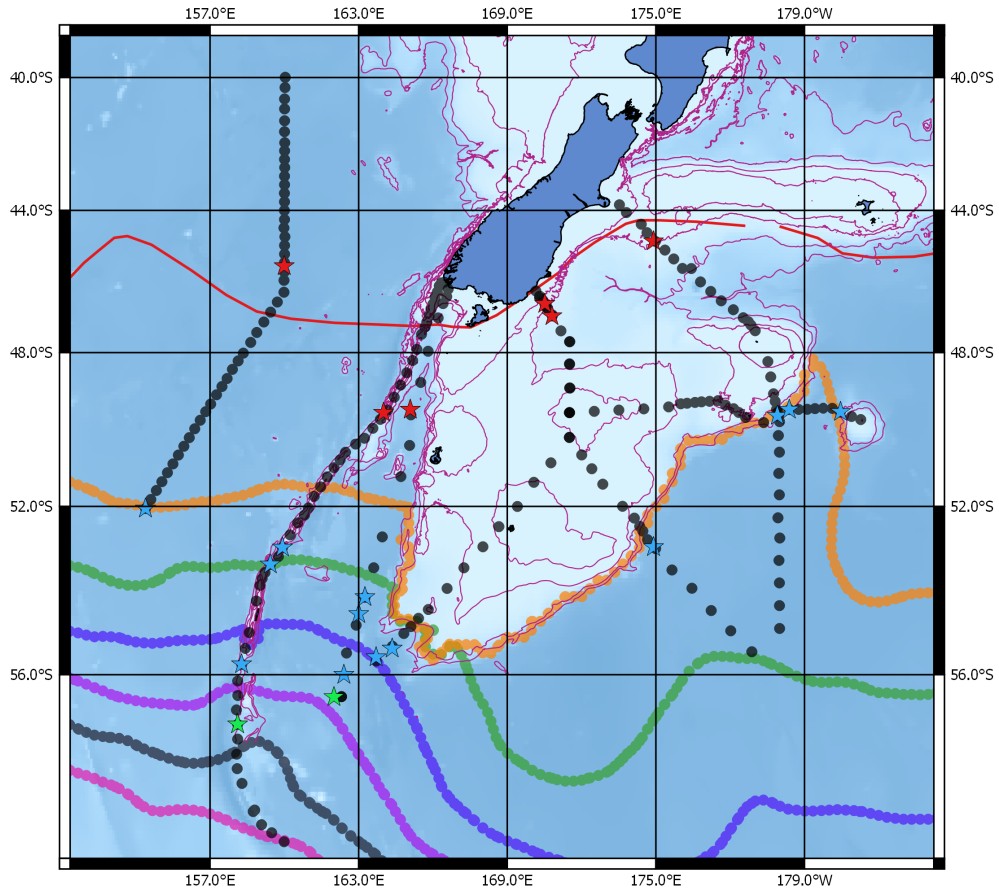

**Figure 9.** STF from Orsi et al. (1995) (red solid line), Southern Ocean fronts distribution referenced to 2500 dbar from Sokolov and Rintoul (2009a), SAF-N (orange), SAF-M (dark green), SAF-S (dark blue), PF-N (purple), PF-M (grey) and PF-S (magenta). The location of the fronts determined in this study (discussed above) are shown by the red (STF), blue (SAF) and green (PF) stars. Bathymetric features indicated on Fig. 1. CTD stations indicated in black.

# 4  Discussion

## 4.1  Fronts

### Subtropical Front

We found that the STF is always further south to that identified by Orsi et al. (1995) except in the Tasman transect where is found north (Fig. 9). This disagreement might come from the fact that Orsi et al. (1995) used significantly less hydrographic data in this region than was used in this study. We found that the STF changes across the region, from being density compensated west of New Zealand to the non-density compensated east of New Zealand. In the Tasman Sea we see a shallow well defined front, but we cannot identify the double structure mentioned by Hamilton (2006) and Smith et al. (2013) in the TAN0609 transect presented here. South of New Zealand the front is density-compensated with no geostrophic flow associated



with it, and it was identified in the same position as Smith et al. (2013), which was further south than previously identified by Butler et al. (1992). However, we agree that there is only one branch of the S-STF in this location due to topographic steering as suggested by Butler et al. (1992) and Smith et al. (2013). East of New Zealand, the front becomes weakly compensated, and it is locally known as the Southland Front (e.g., Heath, 1985). The salinity tongue found at $200\,\mathrm{m}$ that identifies the S-STF

in the Tasman Sea and south of New Zealand is not evident. It has a strong north-east flow close to the continental margin. The Southland Front fits the definition of the dynamical STF found on the western side of the major ocean basins as suggested by Graham and De Boer (2013). This implies, that the Southland Front is different from the STF. This is supported by Sutton (2003), who using the same data as in this study, found that the core of the Southland Current is further off the coast than the Southland Front, and it primarily transports SAW. The implication of this is that STF terminates on the Campbell Plateau

while a dynamical current is generated along the east coast of the South Island. The forcing mechanism of the Southland Current would be much more complex, a combination of large scale winds and bathymetry (e.g., Tilburg et al., 2002). However, variability on the Southland Current is likely initially wind driven as suggested by Chiswell (1996) who found a correlation of local winds to the velocity from moored current-meters.

**Subantarctic Front**

Away from bathymetry three branches in the SAF are expected (e.g., Sokolov and Rintoul, 2002). Bathymetry affects the number of identifiable branches of the SAF. Its role in minimising meandering and stabilising the flow of the SAF as it passes across ridges has been previously suggested by Sokolov and Rintoul (2007), Rintoul et al. (2014) and Chapman and Morrow (2014). It is also consistent with the idea of topographic barriers reducing the number of branches in a front (Thompson

et al., 2010; Graham et al., 2012; Thompson and Sallée, 2012; Chapman, 2014). The SAF-N was identified in the Tasman Sea transect through the geostrophic velocity and transport signal, but was not seen in the hydrography. The other two branches were not crossed as the transect did not extend far enough south. Three branches of the SAF were identified in the Macquarie Ridge transect flowing through the major gaps in the ridge along with the northern branch of the PF at the southern end of the transect. However, in the southwest corner of the Campbell Plateau the three distinct branches were not found. Instead the

hydrographic properties change rapidly suggesting the three branches of the SAF are forced to merge together against the steep flanks of Campbell Plateau (Fig. 5a and b). Further along the Subantarctic Slope the branches diverge with SAF-N following the plateau's bathymetry (Fig. 6a). The middle and southern branches turn sharply to the southeast after $164°E$ - an observation that is consistent with previous studies (Orsi et al., 1995; Belkin and Gordon, 1996; Gille, 2003; Stanton and Morris, 2004; Sokolov and Rintoul, 2007). However, the double flow structure reported by Stanton and Morris (2004) is not found in this

study. One justification for this could be the different methods used on the interpolation of the original data. No interpolation was used in this study.

Near Bollons Seamount the location of the fronts in the hydrographic data agrees with the meander of the SAF-N seen in the SSH (Sokolov and Rintoul, 2009a) (Fig. 9). Similar features were previously described in model simulations by Carter and





Wilkin (1999), who found an eddy structure between 175°W and 170°W and a strong flow at 180°W instead of two strong jets. They suggested the gap between Campbell Plateau and Bollons Seamount was a passage into Bounty Trough. The hydrographic data are unclear, but the SSH do not agree with this hypothesis, suggesting that instead SASW north of Campbell Plateau is most likely sourced through the Pukaki Saddle. Circulation through the Pukaki Saddle is also supported by ALACE

and Argo float data (Davis, 1998, 2005), with float trajectories through the gap and strong flow to the east of Bounty Plateau.

The steep flanks of Campbell Plateau are dominated by the flow associated with the different branches of the SAF. The flow follows the slope for more than $500 \, \mathrm{km}$. The flow separates at Pukaki Saddle, with part flowing north through the gap to the north of the plateau. North of Campbell Plateau and in Bounty Trough the circulation is complex with a recirculation

pattern, probably because of interaction between the flow through Pukaki Saddle and the flow further west associated with the Southland Current.

### 4.2 Water masses

There is a range of surface water masses in this region (Table 3). Neritic Waters (NW) are found close to the coast south of New Zealand and north of the S-STF. Between the S-STF and the SAF we found Subantarctic Surface Water (SASW). In the area

that typically is called "in transition" between the SAF and the PF we found distinct step changes in water properties rather than a continuous change. These step changes between the water masses is associated with the presence of the two fronts that strongly influence the distribution of the different water masses (Fig. 4e and 5e).

Intermediate waters here comprises Subantarctic Mode Water (SAMW) and Antarctic Intermediate Water (AAIW). The pres-

ence or absence of SAMW and where it is formed in this region has been highly debated. We find that SAMW is present in most of the hydrographic transects, except along Macquarie Ridge. The SAMW found on the Tasman transect is slightly different to that on Campbell Plateau. Tasman SAMW is warmer and saltier ($9 \, °\mathrm{C}$, $34.7 \, \mathrm{g \, kg^{-1}}$) than Campbell Plateau's SAMW ($7 \, °\mathrm{C}$, $34.4 \, \mathrm{g \, kg^{-1}}$). As previously noted by Rintoul and Bullister (1999), and could be due to the presence of the bathymetric ridges (e.g., Piola and Georgi, 1982; Aoki et al., 2007), i.e. Macquarie Ridge acting as a barrier between the Tasman SAMW and

Campbell Plateau SAMW. One hypothesis for the absence of SAMW on the ridge is that the complexity of the topography and relatively shallow topography results in mixing and does not allow SAMW to maintain an obvious thermostad which defines the SAMW (Fig. 3).

SAMW over Campbell Plateau has a seasonal signal, forming in winter on the northern side of the SAF as it passes south

of the plateau (Fig. 10). It can be most accurately mapped using potential vorticity $<0.5 \times 10^{-9} \, \mathrm{m^{-1} \, s^{-1}}$, and a potential density anomaly between 26.80 and $27.06 \, \mathrm{kg \, m^{-3}}$ (Hasson et al., 2011; Hartin et al., 2011). The eastern Campbell Plateau transect has been repeated during different seasons and shows the seasonality of the SAMW (Fig. 10). In winter SAMW outcrops on the north side of the SAF and is found over the plateau. In summer the upper $100 \, \mathrm{m}$ of the water column is stratified and SAMW is found below this layer. In autumn SAMW lies between 200 and $500 \, \mathrm{m}$ depth, and appears to be a residual from the previous



winter. This is consistent with the theory that SAMW forms on the northern side of the SAF (e.g., McCartney, 1977; Butler et al., 1992; Morris et al., 2001; Sallée et al., 2006; Hasson et al., 2011). However, Heath (1981, 1985)'s hypothesis of SAMW forming over the Campbell Plateau cannot be excluded.

5  Griffith et al. (2010) reported two types of SAMW on the Pukaki Saddle area. We suggest that there is only one type of SAMW on the Pukaki Rise and that the second type of SAMW identified by Griffith et al. (2010) is more likely to be an eddy-like structure with warm and salty STW, sourced from an eddy generated at the STF (e.g., Williams, 2004).

Southern Ocean AAIW as described by Bostock et al. (2013) is found in different locations in the subantarctic region. In
10  the Tasman Sea AAIW is formed just to the north of the SAF (Fig. 2a and b). East of New Zealand AAIW is found around the edges of Campbell Plateau at a depth between 600 m (Fig. 11b,c,d,f) and 1000 m (Fig. 11a). AAIW also passes through Pukaki Saddle (Fig. 11e). This AAIW is defined as the Southern Ocean type, as it flows in directly from the Southern Ocean, by Bostock et al. (2013), who concluded AAIW flowed around Campbell Plateau and through Pukaki Saddle from float data. Our hydrographic observations are consistent with this interpretation.

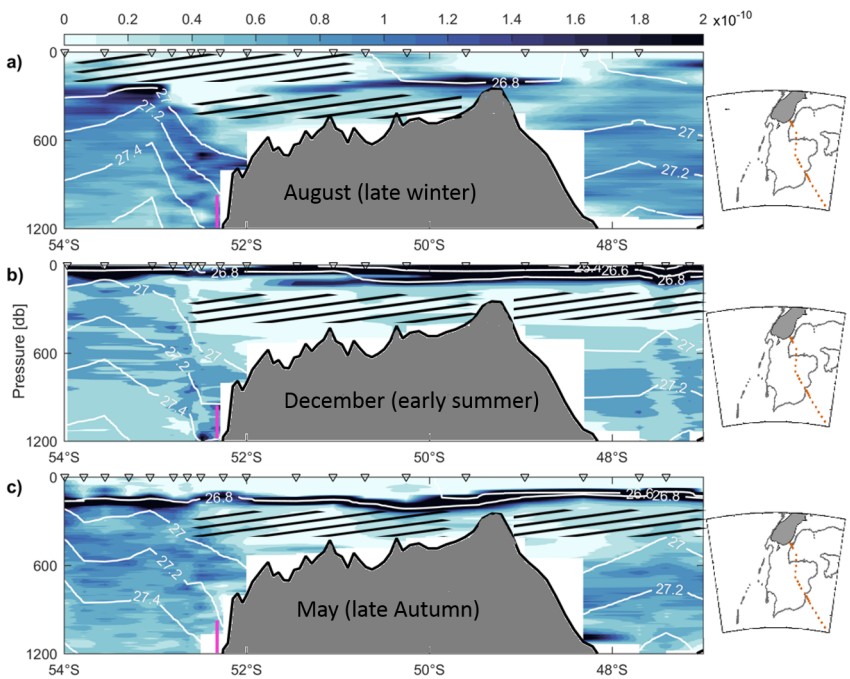

**Figure 10.** Potential Vorticity section across Campbell Plateau, a) August (late winter), b) December (early summer) and c) May (late Autumn). Values used to identify SAMW are indicated in Table 3. Potential density anomalies (white contours $kg\,m^{-3}$). Hatching corresponds to SAMW. Magenta bars are location of the SAF as defined by hydrography.



We find in this region that these water masses (SAMW and AAIW) are geographically separated, with AAIW forming west of Macquarie Ridge and SAMW forming southwest of Campbell Plateau. However, the $1000\,\mathrm{m}$ isobath seems to divide the water mass distribution in the New Zealand subantarctic region, with waters within the $1000\,\mathrm{m}$ isobath characterised as Campbell Plateau waters, and the water deeper than that characterized as Southern Ocean waters. This is consistent with Stanton and

5  Morris (2004) who described the $1000\,\mathrm{m}$ contour as the shallowest depth that "felt the SAF influence".

Deep water masses are common in all the profiles and include Upper Circumpolar Deep Water (UCDW) and Lower Circumpolar Deep Water (LCDP), but are not found on Campbell Plateau, because it is too shallow.





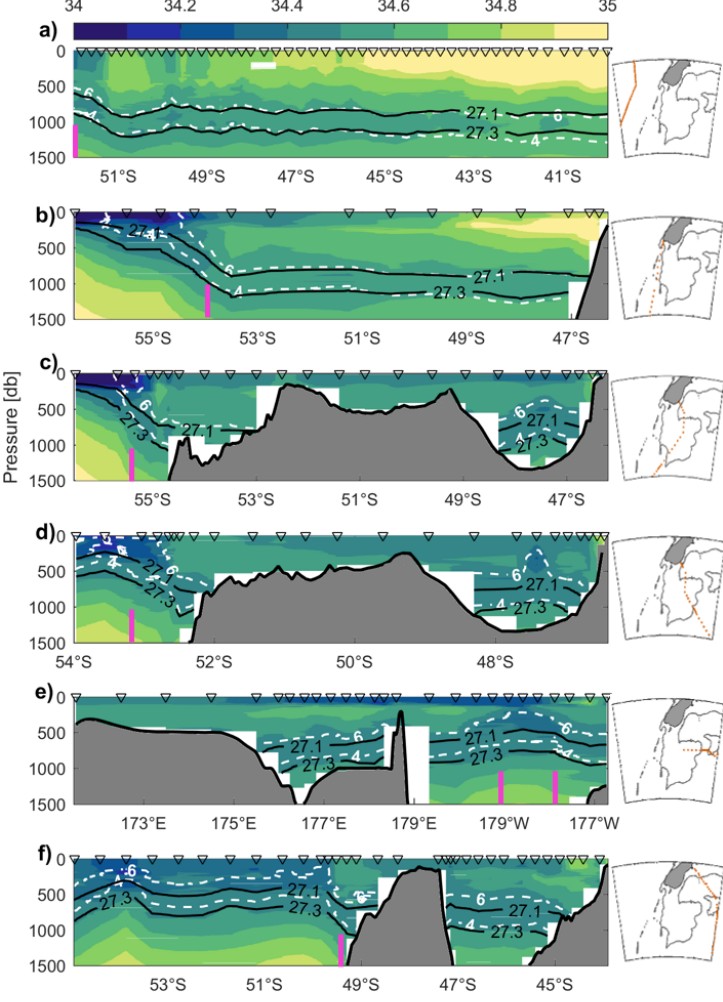

**Figure 11.** Location of AAIW around Campbell Plateau (a) Solander Trough, (b) Western Campbell Plateau, (c) Eastern Campbell Plateau, (d) Eastern and north of Campbell Plateau, (e) Pukaki Saddle and (f) Bounty Trough. Colour code is absolute salinity. Colorbar is absolute salinity. Dashed white lines are conservative temperature, °C, on AAIW range (4-6 °C). Solid lines show the potential density anomaly of the AAIW 27.1 to 27.3 kg m$^{-3}$. Magenta bars are location of the SAF as defined by hydrography.

## 5 Conclusions

For the first time all of the high resolution transects in the subantarctic region of the Southern Ocean south of New Zealand have been integrated to determine the oceanography, fronts and water mass distribution of this unique and complex region of the Southern Ocean. Bathymetry has a dominant role in partitioning and steering the water masses and fronts in the region.

5  Analysis of the water masses showed that the 1000 m isobath divides water mass characterised as Campbell Plateau waters





from the Southern Ocean waters. The 1000 m isobath separates the region into two regimes; open-ocean and a shelf sea. Hence, Campbell Plateau is unique and completely different from other part of the subantarctic with a quite distinctive distribution of water masses very different of the ones around it.

Hydrographic identification of the fronts agrees with SSH definition where bathymetry is not playing a role on the position of the fronts. Reasons for this include: the hydrographic data may not resolve all the branches of fronts, specially where they are close together. There may also be insufficient resolution in the Mean Dynamic Topography (MDT) used to calculate the fronts from SSH particularly when the bathymetry is shallower than the reference depth (2500 m) of the MDT (Sokolov and Rintoul, 2009a) .

The STF around New Zealand is a density compensated front south of New Zealand and a dynamical STF (weakly density compensated) east of New Zealand, namely the Southland Front. Recent research suggest these are actually distinct fronts driven by different mechanisms, suggesting that Campbell Plateau may act as a barrier between the two fronts.

SAMW and AAIW formation are both linked to the SAF (e.g., Hanawa and Talley, 2001; Koshlyakov and Tarakanov, 2005; Hartin et al., 2011). Here we find these water masses (SAMW and AAIW) are geographically separated, with AAIW forming west of Macquarie Ridge and SAMW forming southwest of Campbell Plateau. This distribution suggests that bathymetry plays an important role in the type of intermediate water that forms and thus its distribution in the Southern Ocean. The spatial and temporal distribution of the observations remain insufficient to separate the two hypothesis around SAMW formation, whether
it is formed on or adjacent to Campbell Plateau.

Understanding the variability on Campbell Plateau and resolving the seasonal cycle for mode water formation will require additional observations, particularly during winter and on the western plateau which is poorly resolved in this study. Specifically year-round moorings or similar data and additional sections in the south west are needed to better resolve the SAMW
formation question.

Campbell Plateau's topography geographically locks the STF and the SAF and affects the access of the water masses that form at these fronts on to the plateau. This makes Campbell Plateau's oceanography unique at both subantarctic and global scales. It is unclear how a changing climate may alter the oceanography of this unique region. Understanding this vulnerability
and its implications for diverse marine life that lives on the plateau requires continuous investigation.

*Acknowledgements.* We thank the captains, crews and scientific staff for their work in collecting the data on the many voyages on RV *Tangaroa* funded by the New Zealand government. We thank Phil Sutton for his helpful comments and conversations. This work has been





supported by Victoria University of Wellington Doctoral Scholarship and NIWA Post-Doctoral Fellowship to A. Forcén-Vázquez. Funding for this work came from Antarctic Research Centre Endowed Development Fund (Victoria University Foundation), and NIWA core funding.



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
