# Peer review of "Campbell Plateau: A major control on the SW Pacific sector of the Southern Ocean circulation."

_Ocean Science, 2017_

## Referee Comment (RC1) · Anonymous Referee #1 · 15 Jun 2017

The work describes the water mass properties, oceanic fronts and associated geostrophic circulation of the SW Pacific sector of the Southern Ocean between 40S and 60S based on 8 (or 7) hydrographic transects collected between 1998 and 2008. The hydrographic data have been used in other publication as the author mention but never all together. Despite this work being basically a description of the hydrographic sections, it may become a good review of the circulation, fronts and water mass properties of the New Zeeland Sub-Antarctic region after a mid-to-major revision. I believe the authors should explore more the hydrographic data to give a new and unique contribution to the literature. I will give some suggestions below to improve the quality and readability of the paper.

[Figure]

1) Throughout the MS, the authors talk about compensated fronts on that region. I suggest the authors compute horizontal Turner angles and clearly demonstrate that. This would be new. See, for example, the paper below: Tippins, D., and M. Tomczak (2003), Meridional Turner angles and density compensation in the upper ocean, Ocean Dyn., 53, 332–342

2) Figure 10 shows the PV from three sections across the Campbell Plateau. I suggest the authors explore the PV in the other sections as well and make the PV part of the result.

3) I am not sure, but I think the fronts (at least the major ones) would be clearer if you plot the horizontal gradients. Please verify.

4) Figure 9 must be in the beginning of the work because they show the position of the fronts that is important to the readers for better understanding the paper. It should be cited in the introduction. I can give two possible solutions: (1) Add the front curves to Fig.1 or (2) Make Fig 9 becomes Fig. 1B

5) Because all sections occur during the altimetry era, the authors should use altimetric data to support their study and discussion, especially when the discussion is about currents. This will make the paper become much better.

6) It is ok to divide the introduction in subsections, but the subsection titles must have some meaning. For example: "regional setting" of what ? "Previous work": Introduction will be always about previous works.

7) The colorbar used for the temperature plots throughout the MS is not good to distinguish the features the authors mention. Please, try a different colorbar. Salinity is also not great but we can see some of the features. Because the salinity/temperature colorbars are darkish the overlaid contours must be plotted with light colors so we can see clearly the fronts the authors mention.

Abstract

[Figure]

P1 L#5: The abstract states "using eight high resolution oceanographic sections" but in Table 1 there are only 7 listed and in Figure 1 I counted only 7 sections as well. Please, check it.

Introduction:

P2 L#28: "sections collected between 1998 and 2013" but in P5 L#2-3 "between 1998 and 2008". Which statement is correct? I imagine from Table 1 that it is the last one.

P2 L#29: In some point of the MS, the authors need to say why they chose these specific hydrographic transects. Are there more transects? This is not clear in the text.

P4 L#16: "temperature, salinity and usually (...)".. usually?

P4 L#20-32: The point of this paragraph? It is not clear. I found a bit confusing, may be it could be broken into two paragraphs.

P5 Figure 1: What the dot colors mean? Which bathymetric dataset is plotted? ETOPO-2 ? Please specify.

Data and Methods:

P#5 L#5: Are the CTD salinity measurements calibrated against bottle measurements?

P#5 L#5: What is the accuracy for temperature and salinity measurements in these sections? This is important for the deep ocean.

P#5 L#7: There is no mention about Quality Control procedures. Please mention what you have done, since you have reprocessed from the raw data.

P#5 L#9: A reference is need here. Which work has shown that the difference between practical salinity and absolute salinity is defined is 0.17 g/kg? In the whole water column?

P#5 L#11: Conservative Temperature and Potential Temperature are not the same. The differences are small but they are not the same. See: Intergovernmental Oceano-

graphic Commission, Scientific Committee on Oceanic Research, International Association for the Physical Sciences of the Oceans, The International Thermodynamic Equation of Seawater—2010: Calculation and Use of Thermodynamic Properties, Intergovernmental Oceanographic Commission, Manuals and Guides 56 (UNESCO, 2010).

P#5 L#12-13: I didn't understand: "The transects are discussed from west to east, front identification criteria are shown in Table 2, and their positions are indicated in each transect". Positions of what? Fronts?

P#6: Because there is a Table (Table 2) with this information, there is no need to write the numbers again here. According with the text, or what I understand from it, you are indeed using both temperature and salinity gradients to define the fronts. However, in Table 2 only for the S-STF the salinity criterion is displayed. Did I understand wrong? Anyhow, I suggest you re-write this part and add the salinity criteria to Table 2.

P#7 Table 2: Add degrees Celsius for temperature, for example, 12oC etc. Are temperatures expressed as in situ, potential or conservative temperature here? Does salinity in the S-STF refer to absolute salinity? If yes, please add units. In the PF row: <2 of what? Another question: what the authors mean by further south? For all other fronts in this table there is no mention to geographic distribution.

P#7 L#9: Give at least the order of the "level of no motion". For example, 2000, 3000, 4000 dbar? Would be the level of no motion near the ocean bottom? Are they very different between adjacent pairs? Please discuss the choice for the level of no motion and on the impact in the geostrophic currents.

P#7 L#10: "Cumulative transport was calculated by integrating the transport along the whole water column, setting the zero transport at the stations closer to the coast" For the transect in the Tasman Sea, this description does not make sense because the closest point would be in the middle of the transect around 46S.

P#8 Table 3: Absolute salinity unit is missing. Would be temperature in situ, potential or conservative? Density here would be sigma-0 or neutral density? In the Density column: 1500-3000 (NADW) and 1500 (UCDW) are not density. I think they are in the wrong column. Wouldn't be depth?

Results:

P#8 L#4: define in some point of text that winter is austral winter .

P#8 L#1-5: It is very hard to identify anything near the surface in Figure 2 (even if I make a zoom). Consider plotting the first 500 m in a sep-arated plot such as in the WOCE Atlas. See this example: http://whp-atlas.ucsd.edu/pacific/p14/sections/printatlas/P14_THETA_final.jpg. Also the WOCE colobars make the plots very clear. Try to use something similar in your plots for tem-perature and salinity. Here, you can see several good examples of hydrographic section plots: http://whp-atlas.ucsd.edu/pacific/p14/sections/printatlas/printatlas.htm

P#8 L#8: "The tongue evolves into a weakly stratified layer in the upper 500m of the water column". You could use PV, N2 or a density section to show the weak stratifica-tion.

P#8 L#12: Doubt: Is this section only down to 3000 dbar or it was just the plot. Because if it goes below than 3000 dbar, it is important to say whether or not the LCDW can be identified, which depth range it occupies etc even if not shown. From the T/S diagram, it seems that the LCDW is identified in this transect.

P#9 L#1: Figure 2c seems to indicate the presence of several eddies in the first 1500 m and not continuous recirculations (?). Please expand the comment about that. The authors should use an altimetry map from the cruise time to show the mesoscale eddy activity and also add some information about the ACC here in that time. Is the strong jet of > 25 cm/s associated to the ACC?

P#10 L#6: Use horizontal Turner angles to show that the front is compensated. Please

also look altimetry data to see if the ACC meanders in this region (the section is close to the Campbell Plateau) and eddies in the time of the cruise.

P#12 L#4-5: "The front here is density compensated, hence there is no flow or transport associated with it". Again, use horizontal Turner Angle to clearly show that the front is compensated.

P# 22 "Section 3.2 Hydrographic fronts vs Sea Surface Height fronts". First, this section is already discussion, not results, so it should be in section 4. The authors need to discuss interannual, decadal and seasonal variability here and how this maybe connected with the differences they observed. This is a very dynamical region. For example, in Orsi' paper, the sections are between 1960-90 and in the present paper is between 1998-2008.

Discussion:

P#23 L#5: The authors need to discuss the issue of temporal variability. Hydrographic sections are snapshots of the ocean state.

P#23 L#7-onwards: In my opinion, the authors did not demonstrate that the fronts are compensated. They might be. Please use horizontal Turner Angle or horizontal density ratio to show that. Rudnick, DL, Martin JP. 2002. On the horizontal density ratio in the upper ocean. Dynamics of Atmospheres and Oceans. 36:3-21.

P#29 L#5-9: There are now absolute MDTs without need to use reference level.

---

## Referee Comment (RC2) · Anonymous Referee #2 · 18 Jul 2017

This is manuscript the authors synthesise hydrographic data from 7 transects undertaken over a period of 10 years in the region south of New Zealand. They identify frontal features and water masses. The style suggest that this manuscript is a review, however the authors do not make this clear. There is potential for more work here. As is, I find the work rather thin, since it monotonously describes each transect sequentially with no final integrative synthesis as promised in the abstract/introduction. In addition, the lack of novel information and/or lessons learnt from this exercise lead me to recommend substantial revision. After all, some of the data have already been published.

[Figure]

It is difficult to provide any detailed comments on the current version of the manuscript. The identification and description of fronts and water masses are for the most part fine. The putting this-all-together is missing.

In the introduction, which is rather lengthy, I suggest less textbook-like style and more targeted proses. What are the questions addressed in this work? What can be done with all this data that could not be done before?

What about seasonality - the data has been collected over a range of years and months? Is there any way of also using other datasets (Argo for example) to look at changes over time in this region (perhaps not for Campbell plateau, but the surrounding)?

––––––––––––––––––––––––––––––––

---

## Referee Comment (RC3) · Anonymous Referee #3 · 7 Aug 2017

I was very interested in the synthesis of the series of hydrographic sections around New Zealand. A lot can be learned from revisiting old data in light of new understanding and new observations. However, the work presented doesn't go very deep into the data and is a fairly basic description of the hydrography. It's great to see all of the sections presented but there is no real synthesis in the end.

In the discussion, there is a comparison of the location of the fronts relative to previous studies. This seems the ideal lead into an examination of the interannual and seasonal variability in the front locations using altimetry. Or if Sokolov and Rintoul's (2009) paper did this in sufficient detail then their results could be brought more clearly into the

discussion. Another comparison that could be easily made is to compare the vertical structure of watermass properties and geostrophic velocity along the hydrographic sections with sections constructed from the Argo (or other) climatology using the positions of the CTD stations. In this way the representativeness of the CTD sampling could be assessed, and some comment made about the seasonal and (perhaps) interannual change that is seen in the frontal structure.

I also agree with reviewer 1's excellent and detailed suggestions. Delving more into the density compensation using tools like Turner angles would be quite straight forward, and would add extra depth to the analysis.

I felt that this is on the way to being an excellent review of the circulation south of NZ, but it deserves to be framed better and I recommend the authors spend the time to at least examine the question of variability and representativeness of the sections.

Some detailed comments:

P4, line 16 – missing word, is it "density"?

Fig. 1 - It would be helpful to see the front positions from earlier studies on this map before you describe each of the sections.

P7, line 2 – "SR3 to the east of this region at 140E", should be "west" and "near 140E".

Figures 2-8 – The vertical sections are too compressed and it's difficult to make out the features in temperature and salinity. The black contours are also hard to read on the dark colours. Could you let panels a-d take up most of the page and place panels e and f on the RHS of a-d. Panel d doesn't need as much height as a-c.

P16, line 5 – insert "with" between "associated it"; change "front so is the transport" to "front, as is the transport"

P16, line 10 – "A strong jet >70 cm/s throughout the whole water column" – that's not evident in the figure. I think you mean "A strong jet throughout the whole water column,

with speeds up to 70 cm/s near the surface"

P16, line 11 – AAWS should be AASW.

P23, line 4 – change "where is found north" to "where it is found further north".

P24, line 13 – Change "a correlation of local winds to the velocity" to "a correlation between local winds and the velocity".

P25, line 23 – The sentence beginning "As previously noted . . . " is incomplete and needs to be revised.

P27, line 5 – LCDP should be LCDW.

P29, line 11 – "Recent research" needs a reference.